# Arterial Hypertension and Plasma Glucose Modulate the Vasoactive Effects of Nitroso-Sulfide Coupled Signaling in Human Intrarenal Arteries

**DOI:** 10.3390/molecules25122886

**Published:** 2020-06-23

**Authors:** Sona Cacanyiova, Katarina Krskova, Stefan Zorad, Karel Frimmel, Magdalena Drobna, Zuzana Valaskova, Anton Misak, Samuel Golas, Jan Breza, Jan Breza, Andrea Berenyiova

**Affiliations:** 1Institute of Normal and Pathological Physiology, Center of Experimental Medicine, Slovak Academy of Sciences, 841 04 Bratislava, Slovakia; magdalena.drobna@savba.sk (M.D.); zuzana.valaskova@savba.sk (Z.V.); samuel.golas@savba.sk (S.G.); andrea.berenyiova@savba.sk (A.B.); 2Institute of Experimental Endocrinology, Biomedical Research Center, Slovak Academy of Sciences, 845 05 Bratislava, Slovakia; ueenkrsk@savba.sk (K.K.); stefan.zorad@savba.sk (S.Z.); 3Institute for Heart Research, Center of Experimental Medicine, Slovak Academy of Sciences, 841 04 Bratislava, Slovakia; karel.frimmel@savba.sk; 4Institute of Pathophysiology, Faculty of Medicine, Comenius University, 811 08 Bratislava, Slovakia; 5Institute of Clinical and Translational Research, Biomedical Research Center, Slovak Academy of Sciences, 845 05 Bratislava, Slovakia; anton.misak@savba.sk; 6Department of Urology, Derer’s University Hospital, 833 05 Bratislava, Slovakia; janbreza@gmail.com (J.B.); janbreza@yahoo.com (J.B.J.)

**Keywords:** hydrogen sulfide, S-nitrosoglutathione, human lobar artery, normotensive, arterial hypertension, CBS, CTH gene expression, immunofluorescence, nitroso-sulfide signaling

## Abstract

We have investigated the vasoactive effects of the coupled nitro-sulfide signaling pathway in lobar arteries (LAs) isolated from the nephrectomized kidneys of cancer patients: normotensive patients (NT) and patients with arterial hypertension (AH). LAs of patients with AH revealed endothelial dysfunction, which was associated with an increased response to the exogenous NO donor, nitrosoglutathione (GSNO). The interaction of GSNO with the H_2_S donor triggered a specific vasoactive response. Unlike in normotensive patients, in patients with AH, the starting and returning of the vasorelaxation induced by the end-products of the H_2_S-GSNO interaction (S/GSNO) was significantly faster, however, without the potentiation of the maximum. Moreover, increasing glycemia shortened the time required to reach 50% of the maximum vasorelaxant response induced by S/GSNO products so modulating their final effect. Moreover, we found out that, unlike K+ channel activation, cGMP pathway and HNO as probable mediator could be involved in mechanisms of S/GSNO action. For the first time, we demonstrated the expression of genes coding H_2_S-producing enzymes in perivascular adipose tissue and we showed the localization of these enzymes in LAs of normotensive patients and in patients with AH. Our study confirmed that the heterogeneity of specific nitroso-sulfide vasoactive signaling exists depending on the occurrence of hypertension associated with increased plasma glucose level. Endogenous H_2_S and the end-products of the H_2_S-GSNO interaction could represent prospective pharmacological targets to modulate the vasoactive properties of human intrarenal arteries.

## 1. Introduction

H_2_S is an important gaseous mediator that is presumed to control blood pressure by the direct regulation of vascular tone. Moreover, a deficiency in the H_2_S signaling pathway plays an important role in the development of certain cardiovascular complications, such as spontaneous hypertension, atherosclerosis, heart failure and diabetes [1,2,3,4]. Zhong et al. [4] declared that H_2_S can protect the peripheral vasculature against diabetes-induced injury. In a rat model of streptozotocin-induced diabetes, the authors found a decreased H_2_S concentration in response to high glucose and confirmed that treatment with an H_2_S donor significantly reverses the proliferation of smooth muscle cells caused by hyperglycemia. Therefore, a high concentration of glucose causes a decrease in intracellular glutathione (GSH) [5]. GSH represents a significant part of the cellular antioxidant defense system that could play an important role in maintaining blood pressure in patients with hypertension. Significant disturbances in GSH and GSH-related enzyme levels in elderly patients treated for hypertension have been observed [6]. GSH also plays a significant role in NO metabolism. It has been reported that thiol supplementation with GSH selectively improved human endothelial dysfunction by increasing NO bioavailability [7]. Moreover, S-nitrosothiols (RSNOs) formed from the NO-dependent S-nitrosation of thiol-containing proteins and peptides such as GSH are implicated in the in vivo transport, storage, and metabolism of NO [8]. RSNOs seem to be essential in NO biology and blood pressure regulation. Broniowska et al. [9] declared that the release of NO in the vasculature could also belong to the effects observed with administration of one of the RSNOs, namely, S-nitrosoglutathione (GSNO). These beneficial effects of GSNO administration in the vasculature are related, in addition to other effects, to systemic vasodilation [10]. Moreover, Rassaf et al. [10] demonstrated that intravenous application of GSNO into the human systemic circulation mimicked the vascular effects of NO, whereas nitrite and nitrate were inactive. GSNO, which is well tolerated in humans, remains an important molecule of study and a potential therapeutic agent. Our previous results on the human lobar arteries of patients suffering from arterial hypertension confirmed that H_2_S in interaction with GSNO triggered specific nitroso-sulfide coupled signaling [11]. This effect implied the ability to regulate a vasoconstrictor-induced increase in arterial tone towards a more pronounced vasorelaxation compared to the effect of GSNO alone.

Taken together, disorders in GSH-, GSNO- and H_2_S-related signaling pathways could be associated with the development of several pathological stages, and searching for an effective pharmacologic tool based on the mutual interaction of nitroso- and sulfide donors seems to be actual. In this study, we continued with investigation of the vasoactive effects of the coupled nitroso-sulfide signaling pathway in human lobar arteries of normotensive and hypertensive patients with a focus on the importance of glucose levels. The main aim was to investigate the vasoactive effect of the mixture of the end-products of the H_2_S-GSNO interaction (S/GSNO). We also evaluated the gene expression of enzymes responsible for the endogenous production of H_2_S (cystathionine γ-lyase (CTH) and cystathione β-synthase (CBS)) in the arterial wall and surrounding perivascular adipose tissue and we showed the localization of the mentioned proteins using immunofluorescence.

## 2. Results

### 2.1. Plasma Glucose Level

There was a significant difference in the glucose level determined in the plasma of patients (*n* = 13). The glucose concentration in patients with hypertension was significantly increased (6.97 ± 0.25 mmol/L) compared to normotensive patients (5.76 ± 0.11; *p* < 0.01).

### 2.2. Gene Expression of CTH and CBS (RT-PCR)

ANOVA revealed significantly higher (*p* < 0.001) CBS gene expression in perivascular adipose tissue than in the arterial wall in both normotensive and hypertensive patients (Figure 1). The gene expression of CTH was observed only in perivascular adipose tissue since it was under the detectable limit in the arterial wall of patients. By comparing hypertensive and normotensive patients, we did not notice changes in the mRNA levels of CBS (AH: 2.40 ± 0.87; *n* = 7 vs. NT: 3.55 ± 1.32; *n* = 6) and CTH (AH: 0.55 ± 0.17; *n* = 7 vs. NT: 1.02 ± 0.59; *n* = 6) in perivascular adipose tissue. Similarly, the occurrence of arterial hypertension did not affect the transcription of the CBS gene in the lobar arterial wall (AH: 2.48 ± 0.49; *n* = 7 vs. NT: 2.93 ± 0.43; *n* = 6).

### 2.3. Immunofluorescence Localization of CBS and CTH

Both proteins, CBS and CTH, were showed in arterial wall of lobar arteries of both normotensive patients and patients with arterial hypertension (Figure 2). Both proteins were confirmed in media and adventitia, which is in close contact with perivascular adipose cells (Figure 2a). We did not observe CTH in all cells of vascular wall while CBS was homogenously distributed. In cells where we observed mentioned proteins, CTH was located near the nuclei and CBS loosely in the cytoplasm (Figure 2b).

### 2.4. Vasoactive Responses of Human Lobar Arteries

First, we evaluated the functional properties of the endothelium. Bolus application of acetylcholine (10 µmol/L) induced relaxation of the serotonin-precontracted (1 µmol/L) lobar artery (Figure 3). The vasorelaxant effect of acetylcholine (*n* = 6), which is mediated by endothelium-derived endogenous NO, was significantly reduced in patients with arterial hypertension in comparison with the vasorelaxation observed in normotensive patients (*n* = 6) (*p* < 0.01).

Then, we evaluated the relaxations induced by GSNO and the S/GSNO products over time. We assessed the values of arterial tone in the 1st, 2nd, 5th and 10th minute after drug application. We expressed the absolute vasorelaxant effect of all compounds as a percentage of serotonin-induced precontraction. We also expressed the relative vasorelaxant effect (the speed of vasorelaxation) of all compounds as a percentage of the maximal relaxation. Initially, we tested the vasoactive effect of the Na_2_S bolus application in concentrations used for preparing the mixture (0.5 µmol/L) on serotonin-precontracted arterial rings, and we confirmed almost no effect on vascular tone, the application of Na_2_S induced 0.02 ± 0.01 g and 0.03 ± 0.01 g increase of serotonin-induced tone in normotensive (*n* = 6) and hypertensive (*n* = 6) patients, respectively.

In normotensive patients, the application of 0.5 µmol/L GSNO alone induced vasorelaxation, which quickly developed over time. The S/GSNO products (0.5 µmol/L) triggered a pronounced vasorelaxation that was significantly increased compared to the relaxation induced by GSNO alone (*p* = 5.23 × 10^−4^; *n* = 6). Moreover, although there was no significant effect of interaction between used compound and the time of relaxation, there was confirmed a significant dependence of the vasorelaxation induced by all compounds on time (*p* = 2.73 × 10^−12^) (Figure 4a). Moreover, the expression of relative relaxation revealed a quicker development of vasorelaxation induced by the S/GSNO products compared to vasorelaxation induced by GSNO (*p* = 8.49 × 10^−5^) although there was no significant difference in the return of vasorelaxation in comparison with the vasorelaxation induced by GSNO alone (Figure 4b). There was also confirmed a significant dependence of the vasorelaxation on time (*p* = 6.51 × 10^−27^) as well as a significant effect of interaction between used compound and the time of relaxation (*p* = 2.47 × 10^−4^). In patients with hypertension, the application of the S/GSNO products (0.5 µmol/L) did not trigger more pronounced vasorelaxation compared to the relaxation induced by GSNO alone; however, there was confirmed a significant dependence of the vasorelaxation on time (*p* = 8.91 × 10^−10^) as well as a significant effect of interaction between used compound and the time of relaxation (*p* = 0.0013). 

The expression of relative relaxation revealed a significantly faster development of vasorelaxation in its initiation and in its return (*p* = 0.0021; *n* = 6). There was also confirmed a significant dependence of the vasorelaxation on time (*p* = 8.98 × 10^−14^) as well as a significant effect of interaction between used compound and the time of relaxation (*p* = 4.49 × 10^−4^) (Figure 4c,d).

Subsequently, we focused on the difference between the responses of normotensive and hypertensive arteries. The application of 0.5 µmol/L GSNO induced vasorelaxation of serotonin-precontracted (1 µmol/L) renal arteries isolated from normotensive patients, which slowly developed over time. Application of GSNO (0.5 µmol/L) triggered a pronounced vasorelaxation in renal arteries of patients with hypertension (*n* = 6) that was significantly increased compared to the relaxation induced by GSNO (0.5 µmol/L) (*p* = 0.0068; *n* = 6) in renal arteries of normotensive patients (Figure 5a). There was also confirmed a significant dependence of the vasorelaxation on time (*p* = 1.85 × 10^−6^) as well as a significant effect of interaction between hypertension and the time (*p* = 0.0242) (Figure 5a). 

The expression of relative relaxation revealed that vasorelaxation induced by GSNO was faster in arteries isolated from patients with hypertension compared to the vasorelaxation of vessels isolated from normotensive patients (*p* = 0.0097). There was also confirmed a significant dependence of the vasorelaxation on time (*p* = 1.74 × 10^−21^) (Figure 5b).

The application of 0.5 µmol/L S/GSNO products induced similar vasorelaxation of serotonin-precontracted (1 µmol/L) renal arteries isolated from normotensive (*n* = 6) and hypertensive (*n* = 6) patients. Nevertheless, there was observed a significant dependence of the vasorelaxation on time (*p* = 6.01 × 10^−8^) and a significant effect of interaction between hypertension and the time (*p* = 0.0225); (Figure 6a). However, the expression of relative relaxation revealed a quick development of vasorelaxation induced by the S/GSNO products in vessels isolated from patients with hypertension: the initiation and return of the vasorelaxation were both faster compared to the vasorelaxation induced by S/GSNO products in arteries isolated from normotensive patients (*p* = 0.0011). There was observed a significant dependence of the vasorelaxation on time (*p* = 8.70 × 10^−19^) and a significant effect of interaction between hypertension and the time (*p* = 6.26 × 10^−5^) (Figure 6b).

Additionally, we evaluated the time required to reach 50% relaxation. For evaluation of the time to reach 50% relaxation we used the original curves (Figure 7). Time-dependent relaxation fitted the simple exponential decay function, F = y0 +a*exp(-k*t), where k is the rate constant and for τ 1/2 = ln2/k. The values τ1/2 are presented in the Figure 8. The relaxation effect of the S/GSNO products was significantly faster than that of GSNO in both types of arteries. The speed of relaxation triggered by the S/GSNO products was 1.92 and 1.98 times faster in comparison with relaxation triggered by GSNO in normotensive and hypertensive arteries, respectively. Moreover, the speed of relaxation triggered by the S/GSNO products was significantly higher in hypertensive compared to normotensive arteries.

We also evaluated the relationship between the plasma glucose level of all patients and the maximal relaxation and between the plasma glucose level of all patients and the time required to reach 50% of the maximum relaxation. We confirmed no relationship between plasma glucose and maximal relaxation (Figure 9a,b). We also confirmed that there was no dependency as related to the speed in the case of vasorelaxant effect of GSNO (Figure 10a). On the other hand, the speed of relaxation induced by S/GSNO products positively correlated with the glucose level of patients (r^2^ = 0.7484, *p* = 0.0003; Figure 10b).

### 2.5. Vasoactive Responses of Rat Thoracic Aorta

To investigate possible mechanisms involved in the nitroso-sulfide coupled signaling pathway we performed experiments using rat arterial samples isolated from normotensive Wistar rats (*n* = 8) and SHR (*n* = 8) (the limited number of reactions could be followed on human samples to avoid fatigue of the tissue). We evaluated the involvement of the soluble guanylate cyclase, the K^+^ channels (hyperpolarization) signaling pathways, and nitroxyl (HNO) as a mediator, in relaxation induced by S/GSNO products (Figure 11). There was confirmed a significant dependence of the vasorelaxation on the presence of the soluble guanylate cyclase inhibitor ([1H-[1,2,4]oxadiazolo-[4,3-a]quinoxalin-1-one], 10 µM; ODQ) (*p* = 3.21 × 10^−9^) and a significant effect of inhibitor × strain (*p* = 4.82 × 10^−6^) and inhibitor × time (*p* = 6.32 × 10^−8^) interactions were registered. Incubation with an HNO scavenger, N-acetylcysteine (1 mM; NAC), also significantly modified the S/GSNO evoked vasorelaxation (*p* = 6.43 × 10^−6^) and a significant inhibitor × strain (*p* = 3.06 × 10^−4^) and inhibitor × time (*p* = 3.48 × 10^−6^) interactions were confirmed. To demonstrate the possible role of hyperpolarization due to K^+^ channel activation in the vasorelaxation induced by the S/GSNO products, the aortic rings were incubated with a non-specific K+ channel inhibitor, tetraethylammonium chloride (1 mM; TEA). 

A significant dependence of the vasorelaxation on the presence of TEA (*p* = 5.59 × 10^−4^) whereby a significant effect of inhibitor and strain interaction was observed (*p* = 0.022) was confirmed. The evaluation of the data using Bonferroni post hoc test revealed that the vasorelaxation induced by S/GSNO products was inhibited by ODQ in both strains similarly (* *p* < 0.05; + *p* < 0.05, Figure 11a). As related to the effect of NAC, the significant inhibition of the vasorelaxation was observed in arteries isolated from SHR (+ *p* < 0.05) and the significant difference between both strains was confirmed after the treatment with inhibitor (# *p* < 0.05; Figure 11b). The vasorelaxation induced by S/GSNO products was not significantly affected by the treatment with TEA either in Wistar rats or in SHR (Figure 11c). 

## 3. Discussion

The novel finding of the present study is that the heterogeneity of the vasoactive effects of specific nitroso-sulfide signaling in patients exists depending on the occurrence of hypertension associated with increased plasma glucose levels. Moreover, we confirmed for the first time the expression of the CBS gene in human intrarenal arteries and surrounding perivascular tissue.

During the last two decades, it has become apparent that adipose tissue is an active endocrine organ. Feng et al. [12] were the first to demonstrate that H_2_S is produced by epididymal, perirenal and brown adipose tissue in rats. Both CBS and CTH were identified at the mRNA level in these adipose tissue depots; however, the authors suggested that CTH is the main H_2_S synthase. Perivascular adipose tissue (PVAT) produces several beneficial mediators, and H_2_S is one of the adipose tissue-relaxing factors involved in the regulation of vascular tone [13]. Dysregulation of the H_2_S signaling pathway in PVAT could play an important role in pathological conditions such as arterial hypertension, obesity and type 2 diabetes. Experimental obesity induced by a high-calorie diet revealed a time-dependent effect on H_2_S in PVAT; short- and long-term obesity increased and decreased H_2_S production, respectively [14]. Hyperglycemia has been consistently demonstrated to suppress the CTH-H_2_S pathway in various adipose tissue depots [15]. On the other hand, experimental hypertension induced by constriction of the abdominal aorta upregulated the expression of CTH and increased H_2_S production by periaortic adipose tissue [13]. In the present study, we confirmed significantly higher CBS gene expression in PVAT than in the lobar arterial wall in both normotensive and hypertensive patients. Moreover, the gene expression of the CTH gene was observed in PVAT only. Nevertheless, immunofluorescence assay clearly showed localization and distribution of both enzymes, CTH and CBS, in vascular wall–in media and in adventitia into which parts of perivascular adipose tissue penetrates. The absence of CSE RNA in arterial wall might be explained by extremely high metabolic turnover of the specific mRNA. This hypothesis is supported by the presence of CSE protein only in selected cells of arterial wall. This fact-hypothesis however needs further investigation. Nevertheless, our findings confirmed two important facts: the significant positions of PVAT and the CBS enzyme in the production of H_2_S within the human renal arterial system. To the best of our knowledge, similar findings have not been previously reported for the human arterial system. It seems that H_2_S endogenously produced by PVAT could importantly interfere with arterial tone regulation, and any pathological changes in PVAT developed during arterial hypertension or metabolic disorder could affect the impact of the H_2_S signaling pathway on the vasoactive properties of intrarenal arteries.

In our study, we also evaluated and compared the endothelial function of lobar arteries in normotensive patients and in patients with arterial hypertension. We confirmed that the endothelium-dependent vasorelaxation induced by acetylcholine was significantly lower in arteries isolated from hypertensive patients than in those isolated from normotensive patients. This is in agreement with the results of Perticone et al. [16], who observed that acetylcholine-stimulated forearm blood flow was significantly reduced in patients suffering from arterial hypertension. Arterial hypertension linked to endothelial dysfunction and a defective endothelial L-arginine/NO pathway could also be associated with impaired responsiveness to exogenous NO [17]. On the other hand, Munhoz et al. [18] showed that in spontaneously hypertensive rats (SHRs), the animal model of human essential hypertension, an NO donor, sodium nitroprusside, was a more potent and efficient vasorelaxant that induced a release of a greater amount of NO in the arterial wall than in normotensive rats. In our study, we confirmed that GSNO-induced vasorelaxation was significantly higher in arteries isolated from patients with hypertension. Nitrosothiols can act in a way that is independent of the classical NO-cGMP pathway, such as by controlling the protein function by posttranslational modification [19,20]. Although cGMP-independent mechanism of smooth muscle relaxation induced by GSNO have already been supposed, no change in intracellular cAMP level was confirmed [21] suggesting no role for adenyl cyclase pathway. Nevertheless, GSNO has been shown to induce vasorelaxation almost exclusively via releasing NO [22]. It seems that in human intrarenal arteries of patients with essential hypertension, the decreased endothelial NO efficiency might be compensated for by the increased response to exogenous NO.

Our previous studies using isolated rat and human arteries demonstrated that sulfide signaling may be directly associated with nitroso signaling. We confirmed that low doses of a H_2_S donor potentiated NO release from exogenous GSNO and increased its relaxant effect in normotensive rat thoracic aortae [22]. Moreover, the preincubation of the rat aorta with a low H_2_S concentration led to a heightened relaxation induced by GSNO after washing out of the H_2_S donor from the organ bath, demonstrating that H_2_S may form a nitroso-sulfide complex directly in the arterial tissue [23]. In our next study, we confirmed in normotensive Wistar rat thoracic aorta and mesenteric artery that the mutual interaction of the H_2_S donor Na_2_S with GSNO led to the production of new S/GSNO products, which relaxed phenylephrine-precontracted rings with a more than twofold potency compared with GSNO alone [24]. Moreover, the onset of vasorelaxation of the reaction products prepared in a 10/1 molar ratio of Na_2_S/GSNO implied the ability to potentiate and accelerate the vasorelaxant effect of GSNO alone. Similar results were confirmed in the human lobar arteries of patients suffering from arterial hypertension, where S/GSNO products regulated a vasoconstrictor-induced increase in arterial tone towards a more pronounced and accelerated vasorelaxation than GSNO alone [17]. In this study, in normotensive patients, the vasorelaxation induced by S/GSNO products was significantly higher but without any acceleration compared to the effect induced by GSNO. In patients with arterial hypertension, the starting and returning of the vasorelaxation induced by S/GSNO products was significantly faster compared to normotensive patients, however, without the potentiation of the maximum (Figure 4c,d). These results confirmed that the heterogeneity exists and is related to the vasoactive effects of S/GSNO products in animal and human arterial tissues and that the difference in relaxation speed of S/GSNO products in human intrarenal arteries was dependent on the incidence of hypertension.

During the evaluation of the case history of patients, we noticed an increased level of glucose in the plasma of patients suffering from arterial hypertension compared to normotensive patients. Although Perticone et al. [16] demonstrated no significant differences in fasting glucose values between normotensive and hypertensive subjects, they declared that endothelial dysfunction and insulin resistance may be detected in patients with hypertension, increasing their cardiovascular risk profile. Nevertheless, several studies suggest that a mutual relationship between glucose and endothelial function could exist. Guo et al. [25] incubated rat aorta endothelial cells in a medium with increased concentrations of glucose and confirmed that high glucose stimulated the production of intracellular reactive oxygen species, leading to oxidative injury. Chandra et al. [26] demonstrated that exposure of mouse aortas or endothelial cells to hyperglycemia significantly impaired aortic relaxation due to elevated arginase activity and increased reactive oxygen species formation. We suppose that the endothelial dysfunction observed in the arteries of hypertensive patients could also be associated with a negative impact of increased glycemia. Moreover, Emilova et al. [27] demonstrated that exposure of the rat gracilis artery to high glucose conditions evoked the reversion of H_2_S dilatory influence and converted it into contraction. In our study, we wondered whether a relationship exists between the vasoactive effects of GSNO or S/GSNO and glucose levels. Although we did not follow any significant dependence in respect to maximal relaxation we observed that the time required for relaxation to reach 50% of the maximum response induced by S/GSNO was dependent on the patient’s plasma level of glucose. The higher level of glucose in plasma of patients positively correlated with the speed of the development of vasorelaxation induced by the S/GSNO products. Based on these results, we suppose that hypertension associated with increasing glycemia participated in the acceleration of the vasorelaxant effect of S/GSNO products, modulating their final effect.

Molecular mechanism by which S/GSNO forced the vasorelaxant effect of GSNO might have several explanations. Xiao et al. [28] showed that exogenous H_2_S was able to inhibit reactive oxygen species production and to suppress vascular oxidative stress in hypertensive animals so increasing NO bioavailability. Bucci et al. [29] demonstrated that H_2_S could induce vasorelaxation by acting as a nonselective endogenous phosphodiesterase inhibitor that boosts cyclic nucleotide levels in tissues. Authors confirmed that exposure of smooth muscle cells to H_2_S increased their intracellular cGMP content, the product of soluble gunylate cyclase activation induced by NO. Moreover, several new formed compounds have been suggested to mediate the bioactivity of the interaction between nitrosothiols and H_2_S. Cortese-Krott et al. [30] proposed SSNO- to be a major longer-lived product. Filipovic et al. [31] and Wedmann et al. [32] suggested that SSNO- salts could be generators of HNO, and Nava et al. [33] supposed that the ´cross-talk’ observed between H_2_S and NO could be mediated by HSNO and manifested as HNO. In our previous studies, we suggested that the involvement of more than one product was involved in the interaction between GSNO and H_2_S. To evaluate the importance of NO in relaxation effects of GSNO and S/GSNO products, we tested the effect of the NO scavenger, cPTIO [24] in normotensive rats. We confirmed that the relaxation effect of GSNO was effectively inhibited by cPTIO (100 μmol/L) indicating that GSNO acted almost exclusively via releasing NO. On the contrary, relaxation induced by the S/GSNO products was not significantly affected by the presence of cPTIO, indicating that a portion of the relaxation induced by reaction products was mediated by an alternative mechanism, which might directly activate soluble guanylate cyclase without releasing free NO. Moreover, in our recent study we recorded that the hyperpolarization of smooth muscle cells after K^+^-channel activation played no significant role in S/GSNO product-induced relaxation in normotensive rats [34]. In the conditions of arterial hypertension, using renal arteries isolated from spontaneously hypertensive rats (SHR) we found out that the pre-treatment with ODQ, inhibitor of soluble guanylate cyclase, significantly inhibited the vasorelaxant effect induced by the S/GSNO products so indicating the involvement of the cGMP activation in the pathway. Moreover, we observed, that after the pre-treatment with NAC, the HNO scavenger, the development of vasorelaxant response in the time was significantly inhibited [11]. In this study we confirmed that: (a) cGMP pathway was importantly engaged in vasorelaxant effect of S/GSNO in both, normotensive rats and SHR; (b) HNO as a possible mediator of S/GSNO induced vasorelaxation revealed inhibitory effect in conditions of arterial hypertension; and (c) hyperpolarization induced by K+ channel activation was generally not participated in S/GSNO action. Andrews et al. [35] explored the efficacy of HNO, the reduced congener of NO in human blood vessels, and described that it is a vasodilator for humans that is not susceptible to tolerance. The HNO donor caused vasorelaxation of the human saphenous vein and radial artery via activation of a cGMP-dependent pathway; however, unlike glyceryl trinitrate, it did not develop tolerance. Additionally, Sarr et al. [36] declared that GSNO can promote the formation of releasable NO stores in arteries exhibiting increased superoxide levels, whereby increased oxidative stress has been generally demonstrated in conditions of arterial hypertension [37,38]. Taken together, it seems that both GSNO and HNO (as one of potential end-products of the GSNO and H_2_S interaction) could be of potential interest for restoring the protective effect of NO in blood vessels and particularly useful in preventing endothelial dysfunction. Moreover, based on the results of this study, it seems that the modulation of the vasoactive properties of GSNO by reaction with a H_2_S donor probably alleviates the negative effects, such as increased sensitivity and reaction to GSNO as NO donor. We are conscious that there are some limitations of this study. We cannot exclude the possible effect of inflammatory mediators produced by tumor and/or immune cells on vascular responses in comparison with non-cancer patients. The association of tumor with inflammation has already been confirmed, in particular, the development of renal cell carcinoma has been linked to tissue inflammation [39]. During inflammation, initial waves of pro-inflammatory cytokines can stimulate endothelial cells to upregulate adhesion molecules and cytokines that together attract additional immune cells [40]. Moreover, cytokines may have additional specific effects, which could affect the vascular tone and the signaling pathways of vasoconstriction and vasodilation, vascular cell growth and proliferation, and could lead to structural changes in the vessel wall architecture [41]. Nevertheless, although, the effects of nitroso-sulfide coupled signal pathway requires further investigation, the end-products of the H_2_S and GSNO interaction could represent prospective pharmacological tool in arterial hypertension.

## 4. Materials and Methods

### 4.1. Patients

Thirteen patients (10 males, 3 females, age 58.85 ± 3.03) with renal cell carcinoma, a renal pelvic tumor, or a ureteral tumor, who were selected for this study underwent radical nephrectomy. The patients were divided into two groups: seven normotensive patients without diagnosed arterial hypertension or other metabolic disorders and six patients with diagnosed arterial hypertension. The study was conducted under the ethical approval of the Ethics Committee of Derer´s University Hospital and by the Committee on the Ethics of Procedures in Animal, Clinical and other Biomedical Experiments of the Institute of Normal and Pathological Physiology, Centre of Experimental Medicine, Slovak Academy of Sciences in Bratislava (permit number: EK/noh2s/14) in agreement with the Ethical guidelines of the Declaration of Helsinki as revised in 2000. Evaluation of the patient history was performed, and the plasma glucose concentration, determined in the hospital laboratory Synlab using fasting venous blood samples, was registered.

### 4.2. Functional Study

#### 4.2.1. Human Lobar Arteries

Ex vivo experiments were performed on specimens of lobar arteries, specifically branches of the segmental arteries that are branches of the renal artery. All the specimens used for the functional study appeared macroscopically normal without signs of a tumor or inflammation. Immediately after surgical removal, the lobar arteries were kept in Krebs solution of the following composition: 118 mmol/L NaCl, 5 mmol/L KCl, 25 mmol/L NaHCO_3_, 1.2 mmol/L MgSO_4_, 1.2 mmol/L KH_2_PO_4_, 2.5 mmol/L CaCl_2_, 11 mmol/L glucose, and 0.032 mmol/L CaNa_2_EDTA. Isolated lobar arteries were cleaned of surrounding connective tissue and cut into rings (5 mm in length). The rings were vertically fixed between two stainless wire triangles in a 20-mL organ bath with Krebs solution that was oxygenated with 95% O_2_ and 5% CO_2_ and incubated at 37 °C. The upper triangles were connected to sensors of isometric tension (FSG-01, MDE, Budapest, Hungary), and the changes in tension were registered by an AD converter NI USB-6221 (National Instruments, Austin, TX, USA) and DEWESoft (Dewetron, Prague, Czech Republic). The resting tension was adjusted to 1.5 g and applied to each ring. Subsequently, the preparations were allowed to equilibrate for 60–90 min until stress relaxation no longer occurred.

KCl (125 mmol/L, physiological Krebs solution changed to a solution in which NaCl was exchanged for an equimolar concentration of KCl) was added to the organ bath for only 2 min to confirm the sufficient contractility of the sample. The relaxant responses were followed on serotonin (1 µmol/L)-precontracted arterial rings after the achievement of their stable plateau. The rings were then exposed to the maximal concentration of acetylcholine (10 μmol/L) to evaluate the function of the endothelium. We decided not to add concentration-response curves to the experimental protocol to avoid fatigue of the tissue.

Na_2_S·9H_2_O was used as a H_2_S donor that dissociates in a water solution to Na^+^ and S^2−^, which reacts with H^+^ to yield HS^−^ and H_2_S. We used the term Na_2_S to encompass the total mixture of H_2_S, HS^−^ and S^2−^. The stock solution of Na_2_S (100 mmol/L) was prepared by dissolving it in ultrapure deionized water (≥18 MΩ.cm) (Millipore, Darmstadt, Germany) and placing into a −80 °C freezer. On the day of the experiment, the stock solution (100 µl) was thawed and mixed with the buffer: Tris-HCl (200 mmol/L), pH 7.4. This solution was always prepared fresh immediately before the experiment and kept in sealed vials with minimal headspace and used immediately. In the experiment, Na_2_S was applied to the organ bath containing serotonin-precontracted rings at a concentration of 0.5 μmol/L to test that it is not a concentration that induces a significant change in arterial tone. In our previous study, we confirmed that in human lobar arteries, the Na_2_S bolus application at a 1 μmol/L concentration had a minimal effect on vascular tone, and a 20 μmol/L concentration evoked approximately 14% vasorelaxation [11].

The stock solution of S-nitrosoglutathione (GSNO, 10 mmol/L) was prepared by dissolving GSNO in buffer (Tris-HCl (200 mmol/L) and diethylenetriaminepentaacetic acid DTPA (0.1 mmol/L), with a pH adjusted to a value of 7.4) and placing in a −80 °C freezer until use. The concentration of stock solution was also checked spectrophotometrically using an extinction coefficient of 922 M-1 cm-1 at 335 nm. On the day of the experiment, the stock solution (30 µL) was thawed and mixed with buffer (Tris-HCl (200 mmol/L), with a pH adjusted to a value of 7.4) and kept in sealed vials. GSNO was applied to the isolated vessels at a 0.5 μmol/L concentration on serotonin-precontracted rings. The products of the S/GSNO interaction were prepared by mixing both donors, GSNO and Na_2_S, to reach a 10:1 molar excess of Na_2_S over GSNO, which ensures the production of new effective products. Their formation was followed by UV–VIS spectroscopy (absorbance increases at λmax 412 nm) (in detail, also see [23,25]). In short, we mixed 1 mmol/L GSNO with 10 mmol/L Na_2_S at 21 ± 2 °C and waited 3 min until complete formation of the reaction products. To the organ baths, we administered the following final concentration: 0.5 µmol/L S/GSNO = 5 µmol/L Na_2_S + 0.5 µmol/L GSNO. The vasorelaxant responses are expressed as a percentage of the serotonin-induced contraction: the increase in the vascular tone was set as 100%, and the relaxant effect of selected compounds is given relative to this increase. The relative vasorelaxant response (revealing the speed of the vasorelaxation) induced by the mixtures of donors is expressed as a percentage of the maximal relaxation. The time required to reach 50% relaxation and the relationship between the plasma glucose level of all patients and the time required to reach 50% of the maximum relaxation were also determined using linear regression analysis.

#### 4.2.2. Rat Thoracic Aorta

Procedures were performed in accordance with institutional guidelines and were approved by the State Veterinary and Food Administration of the Slovak Republic, by the Committee on the Ethics of Procedures in Animal, Clinical and others Biomedical Experiments (Permit Number: EC/CEM/2017/4) of the Centre of Experimental Medicine and by an Ethical committee according to the European Convention for the Protection of Vertebrate Animals used for Experimental and other Scientific Purposes, Directive 2010/63/EU of the European Parliament. All rats were housed under a 12 h light-12 h dark cycle, at a constant humidity (45–65%) and temperature (20–22 °C), with free access to standard laboratory rat chow and drinking water. The Institute of Normal and Pathological Physiology provided veterinary care. Adult male normotensive Wistar rats (*n* = 8) and spontaneously hypertensive rats (SHR, *n* = 8) were used in this study. Rats were killed by decapitation after a brief anesthetization with CO_2_, and the thoracic aorta was isolated. The artery was cleaned of connective tissue and cut into 5 mm length rings. The rings of vessel were processed using the same methodology as for human lobar arteries. The resting tension was adjusted to 1 g and applied to each ring. KCl (125 mmol/L, physiological Krebs solution changed to a solution in which NaCl was exchanged for an equimolar concentration of KCl) was added to the organ bath for only 2 min to confirm the sufficient contractility of the sample. The products of the S/GSNO interaction were prepared as described above and the final concentrations of S/GSNO were applied after 35 min normalization on noradrenaline (1 µmol/L) pre-contracted arteries. To confirm the involvement of the NO/sGC signaling pathway in the relaxation effect of the S/GSNO products, the sGC inhibitor 1*H*-[1,2,4]oxadiazolo[4,3-a]quinoxalin-1-one (ODQ, 10 µmol/L) was applied 20 min before the addition of the contractile agonist. To test the assumption that vasorelaxation of the mixture is mediated by the release of nitroxyl the experiments were carried out also after 20 min pretreatment with the HNO scavenger, N-acetylcysteine (NAC, 1 mmol/L). To test the involvement of K^+^ channels in the vasorelaxation, the non-selective K^+^ channel inhibitor, tetraethylammonium chloride (TEA, 1 mmol/L) was applied 20 min before the addition of noradrenaline. The absolute vasorelaxant responses were expressed as a percentage of the noradrenaline induced contraction.

### 4.3. RNA Isolation and Real-Time PCR

Total RNA was isolated from frozen lobar arteries and perivascular adipose tissue by the RNeasy Universal Plus Mini Kit (Qiagen, Valencia, CA, USA) and reverse transcribed using the Maxima First Strand cDNA Synthesis Kit (Thermo Fisher Scientific, Waltham, MA, USA). Real-time qPCR was performed on a QuantStudio™ 5 Real-Time PCR System (Applied Biosystems, Thermo Fisher Scientific) using FastStart Universal SYBR Green Master Mix (Roche, Indianapolis, IN, USA). Gene expression was measured using a specific KiCqStart^®^ SYBR^®^ Green Primers Kit (Sigma-Aldrich, St. Louis, MO, USA) for CBS (NM_000071) and CTH (NM_001190463). The obtained data were normalized to the expression of the endogenous reference gene β-actin.

### 4.4. Immunofluorescent Detection of CBS and CTH

Indirect immunofluorescent detection of CTH and CBS was performed on frozen cryostat sections (10 um) of the unfixed LOs of all experimental groups. Series of the sections were fixed in ice-cold methanol for 15 min, blocked by 10% bovine serum with 3% Bovine serum albumin powder for 90 min. Subsequently, sections were incubated with mix of primary antibodies: monoclonal antibody mouse anti-CTH (in dilution 1:200250, Proteintech, Rosemont, IL, USA) and rabbit polyclonal anti-CBS (in dilution 1:250, Proteintech) over night at 4 °C. The sections were subsequently 4-times washed with phosphate-buffered saline (PBS) and then followed the application mix of secondary donkey anti-mouse conjugated with FITC (in dilution 1:500, Jackson ImmunoResearch Laboratory, Inc., Ely, Cambridgeshire, UK) goat anti-rabbit conjugated with Alexa-594 (in dilution 1:500, Jackson ImmunoResearch Laboratory, Inc.). The primary antibodies were omitted in negative controls to check the specificity of immunolabeling. Nuclei was visualized with DAPI (4′,6-diamidino-2-phenylindole) staining during last washing phase. After washing, the tissue sections were mounted into Vectashield mounting medium (Vector Laboratories, Burlingame, CA, USA) and viewed by an A1R+ confocal microscope (Nikon, Tokyo, Japan).

### 4.5. Statistical Analysis

For the statistical evaluation of the differences between groups, one-way, two-way or three-way (rat samples) analysis of variance (ANOVA) was used, followed by the Bonferroni post hoc test. Differences between means were considered significant at *p* < 0.05. All data arising from this study are contained within the article, and any additional data sharing will be considered by the first author upon request.

### 4.6. Drugs

The following drugs were used: KCl, serotonin, acetylcholine, sodium sulfide nonahydrate, S-nitrosoglutathione, Tris(hydroxymethyl)aminomethane, and hydrochloric acid (all from Sigma-Aldrich).

## 5. Conclusions

Our study confirmed that the heterogeneity of specific nitroso-sulfide vasoactive signaling exists depending on the occurrence of hypertension associated with increased plasma glucose levels. We also showed that, unlike K^+^ channel activation, cGMP pathway and HNO as probable mediator could be involved in mechanisms of S/GSNO action. Moreover, we also demonstrated the endogenous production of H_2_S in perivascular adipose tissue and arterial wall of lobar arteries in all patients. From this point of view, we assume that the end-products of the H_2_S-GSNO interaction as well as endogenous sulfide signaling could represent a prospective pharmacological target to modulate the vasoactive properties of human intrarenal arteries.

## Figures and Tables

**Figure 1 molecules-25-02886-f001:**
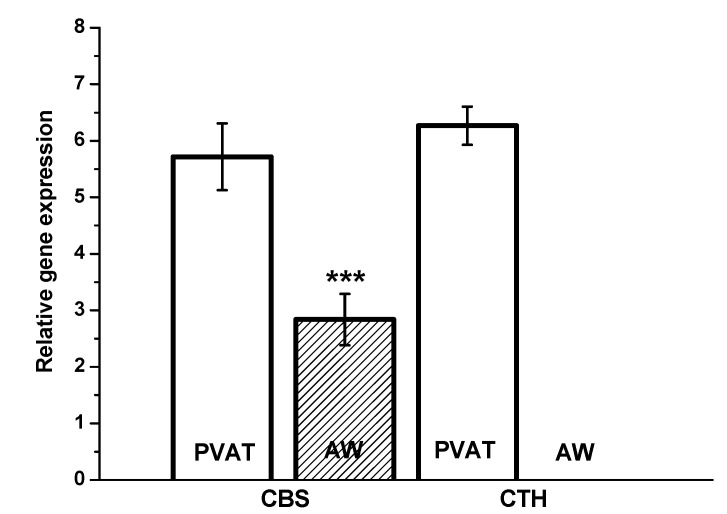
Gene expression of cystathionine beta-synthase (CBS) and cystathionine gamma-synthase (CTH) in perivascular adipose tissue (PVAT) and lobar arterial wall (AW) of all patients. The expression of genes was determined by real-time PCR. The obtained data were normalized to endogenous reference genes. The results are presented as the mean ± S.E.M, and differences between tissues were analyzed by one-way ANOVA with Bonferroni post hoc test on ranks. *** *p* < 0.001 with respect to the CBS expression in PVAT.

**Figure 2 molecules-25-02886-f002:**
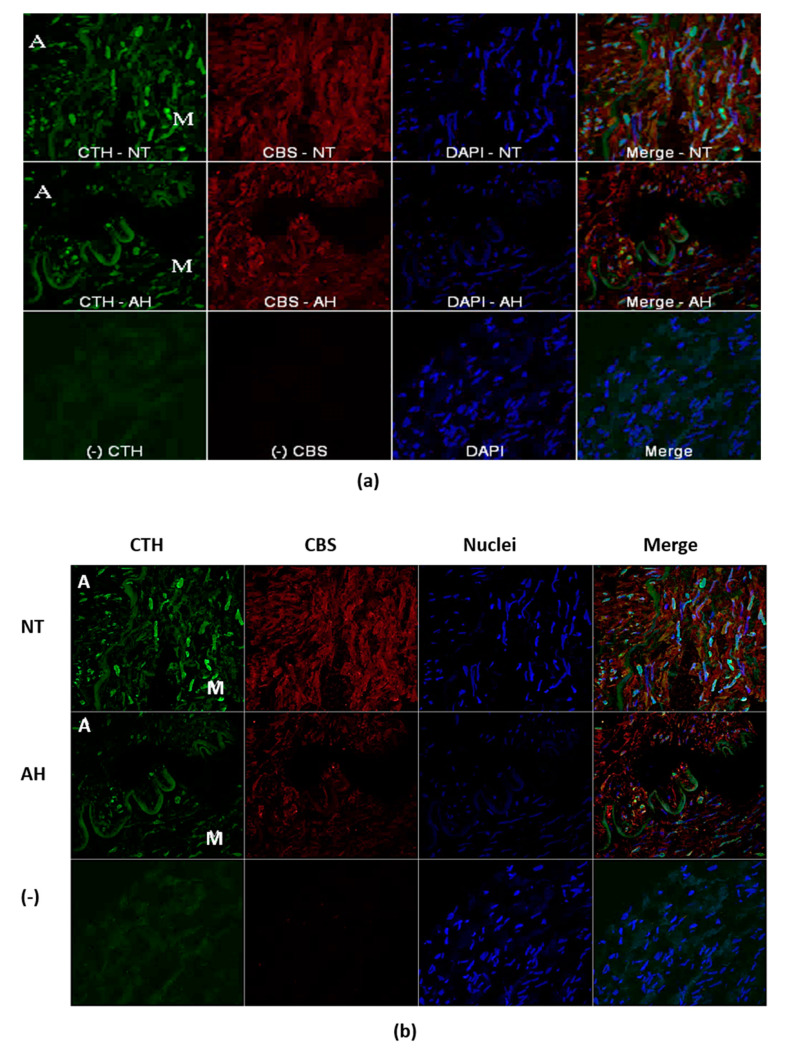
Representative immunofluorescence images in samples of normotensive patients (NT) and patients with arterial hypertension (AH) in arterial wall (**a**). Immunofluorescence localization of cystathionine beta-synthase (CBS-red), cystathionine gamma-synthase (CTH-green) and nuclei (blue) (**b**). M-media, A-adventitia, DAPI-4′,6-diamidino-2-phenylindole, a fluorescent stain. (−)-represents omitted primary antibodies.

**Figure 3 molecules-25-02886-f003:**
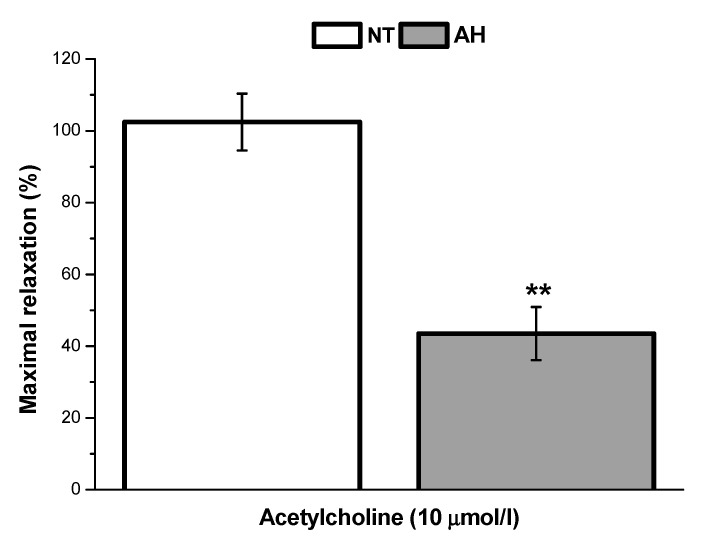
Maximal vasorelaxant responses of serotonin (1 µmol/L)-precontracted human lobar arteries induced by acetylcholine (10 µmol/L) in normotensive patients (NT) and patients with arterial hypertension (AH). Values are the mean ± S.E.M. Significant differences were evaluated by one-way ANOVA. Bonferroni post hoc test was used to describe the differences in mean values of the experimental groups. ** *p* < 0.01 with respect to the NT response.

**Figure 4 molecules-25-02886-f004:**
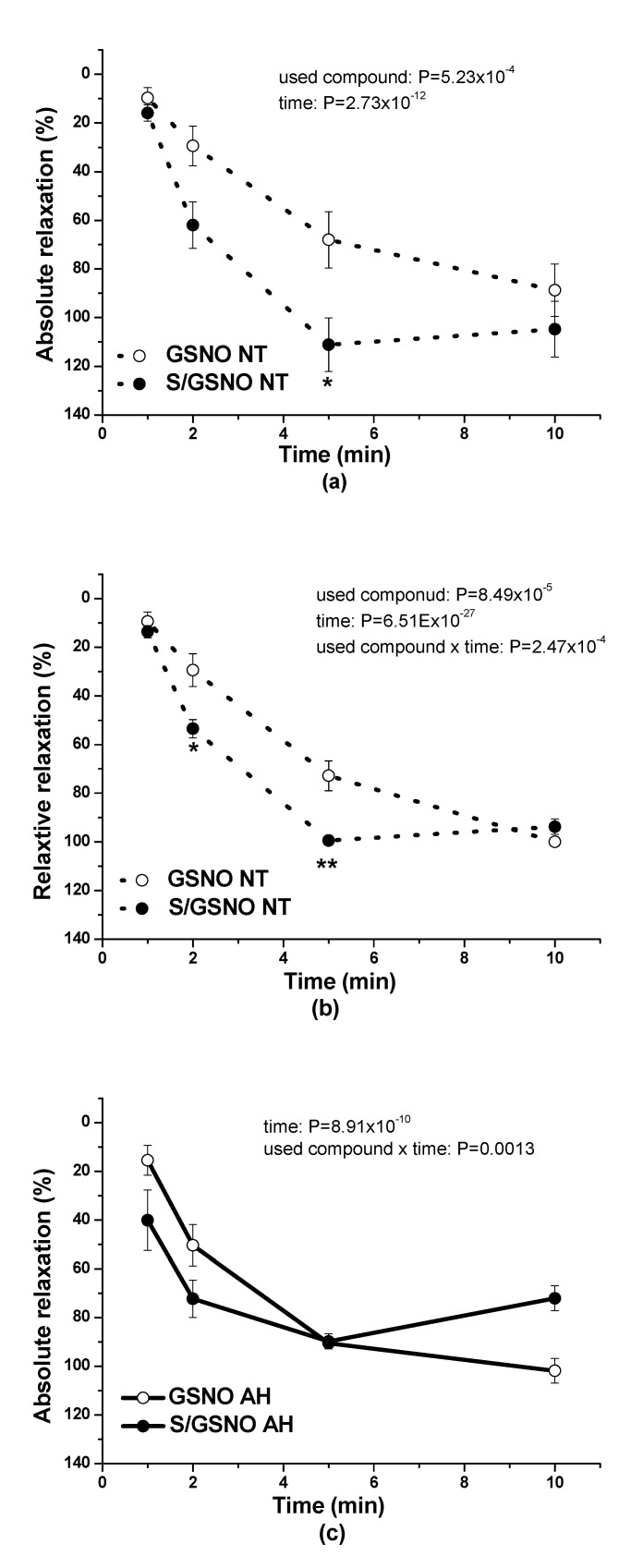
The comparison of vasorelaxations of human lobar arteries induced by GSNO (0.5 µmol/L) and by the S/GSNO products (0.5 µmol/L). The absolute vasorelaxant responses were expressed as a percentage of the serotonin (1 µmol/L)-induced contraction in normotensive patients (NT) (**a**) and in patients with arterial hypertension (AH) (**c**). The relative vasorelaxant responses (revealing the speed of the vasorelaxation) are expressed as a percentage of the maximal relaxation in normotensive patients (NT) (**b**) and in patients with arterial hypertension (AH) (**d**). Results are expressed as the mean ± S.E.M. Significant differences were evaluated by two-way ANOVA for main factors: used compound and time. Bonferroni post hoc test was used to describe the differences in mean values of the experimental groups. * *p* < 0.05; ** *p* < 0.01 with respect to the value of the GSNO response within the appropriate graph.

**Figure 5 molecules-25-02886-f005:**
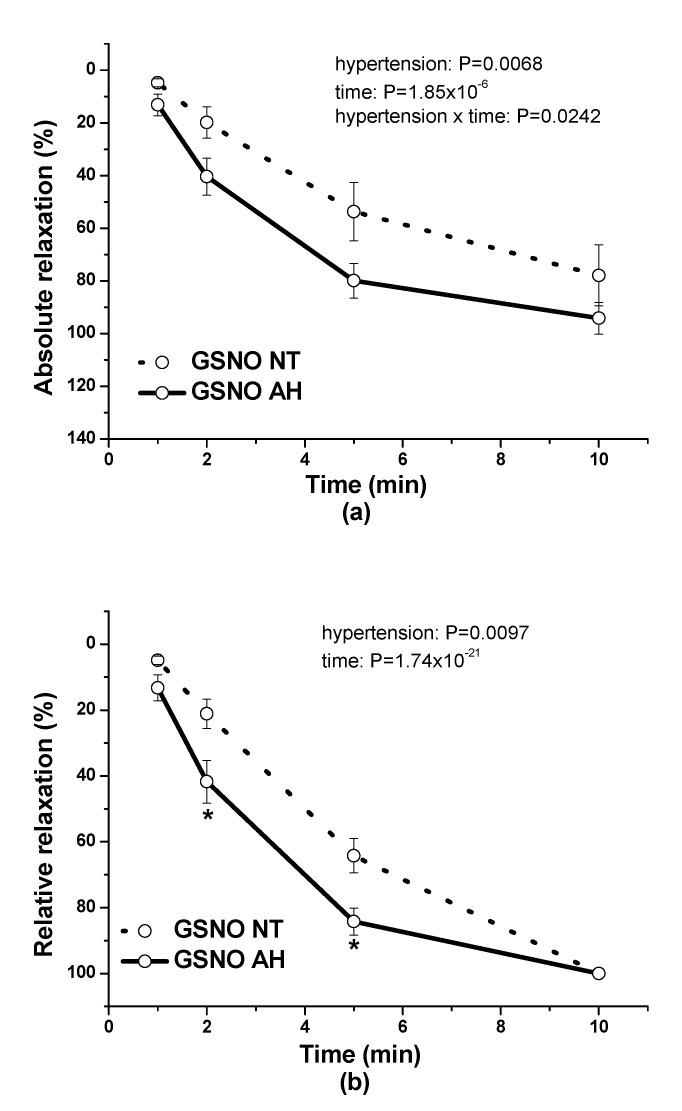
The comparison of vasorelaxations of human lobar arteries induced by GSNO (0.5 µmol/L) between normotensive patients (NT) and patients with arterial hypertension (AH). The absolute vasorelaxant responses are expressed as a percentage of the serotonin (1 µmol/L)-induced contraction (**a**) The relative vasorelaxant responses (revealing the speed of the vasorelaxation) are expressed as a percentage of the maximal relaxation (**b**). Results are expressed as the mean ± S.E.M. Significant differences were evaluated by two-way ANOVA for main factors: hypertension and time. Bonferroni post hoc test was used to describe the differences in mean values of the experimental groups. * *p* < 0.05 with respect to the value of the GSNO NT response.

**Figure 6 molecules-25-02886-f006:**
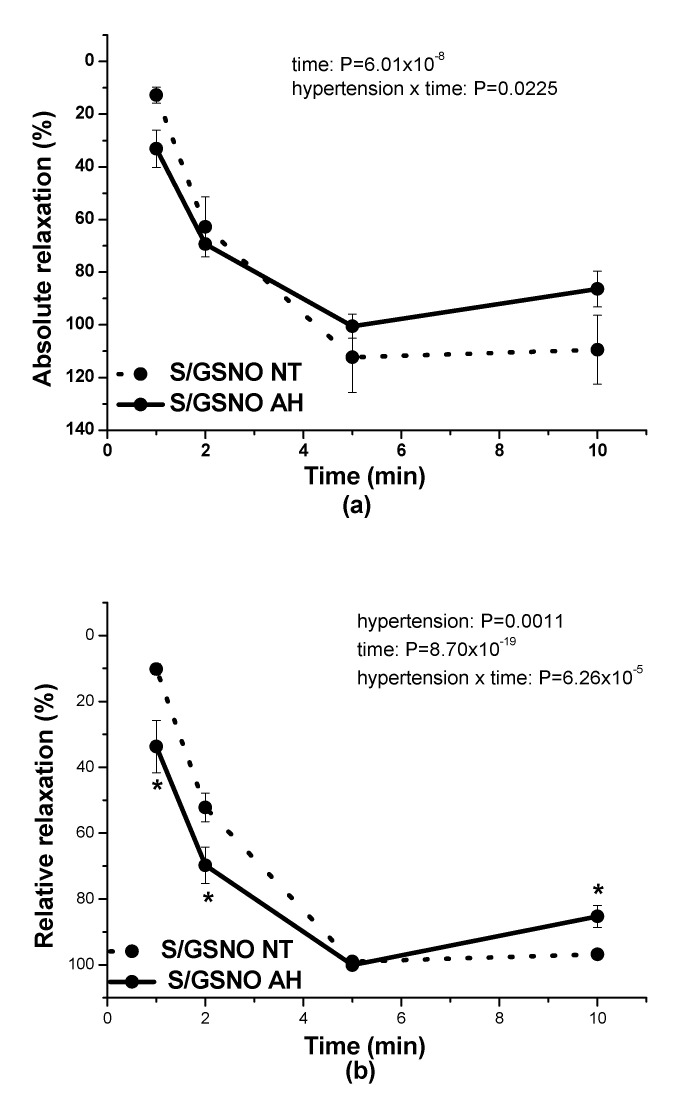
The comparison of vasorelaxations of human lobar arteries induced by S/GSNO (0.5 µmol/L) products between normotensive patients (NT) and patients with arterial hypertension (AH). The absolute vasorelaxant responses are expressed as a percentage of the serotonin (1 µmol/L)-induced contraction (**a**) The relative vasorelaxant responses (revealing the speed of the vasorelaxation) are expressed as a percentage of the maximal relaxation (**b**). Results are expressed as the mean ± S.E.M. Significant differences were evaluated by two-way ANOVA for main factors: hypertension and time. Bonferroni post hoc test was used to describe the differences in mean values of the experimental groups. * *p* < 0.05 with respect to the value of the S/GSNO NT response.

**Figure 7 molecules-25-02886-f007:**
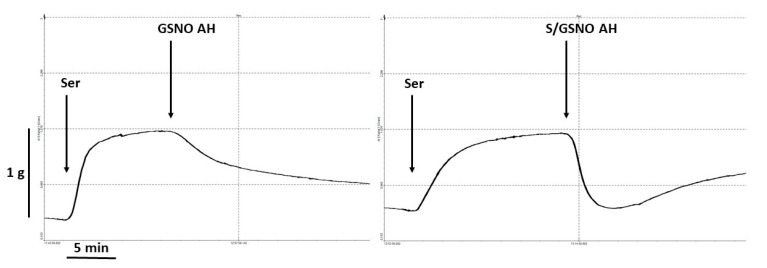
Original record of vasoactive responses induced by GSNO (0.5 µmol/L) and S/GSNO (0.5 µmol/L) products in serotonin (Ser, 1 µmol/L)-precontracted lobar artery of patient with arterial hypertension (AH).

**Figure 8 molecules-25-02886-f008:**
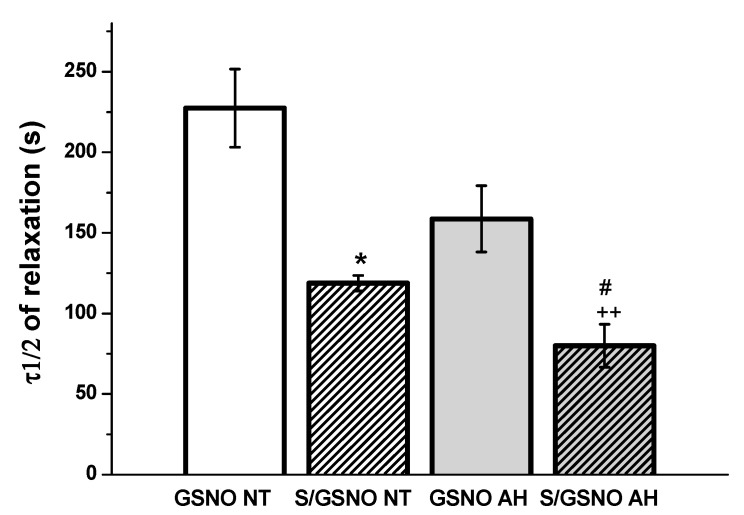
The time required to reach 50% relaxation (τ1/2) induced by GSNO (0.5 µmol/L) and S/GSNO (0.5 µmol/L) products in human lobar arteries of normotensive patients (NT) and patients with arterial hypertension (AH). Values are the mean ± S.E.M. Significant differences were evaluated by one-way ANOVA. Bonferroni post hoc test was used to describe the differences in mean values of the experimental groups. * *p* < 0.05 with respect to the value of the GSNO NT response, ++ *p* < 0.01 with respect to the value of the S/GSNO AH response, # *p* < 0.05 with respect to the value of the S/GSNO NT response.

**Figure 9 molecules-25-02886-f009:**
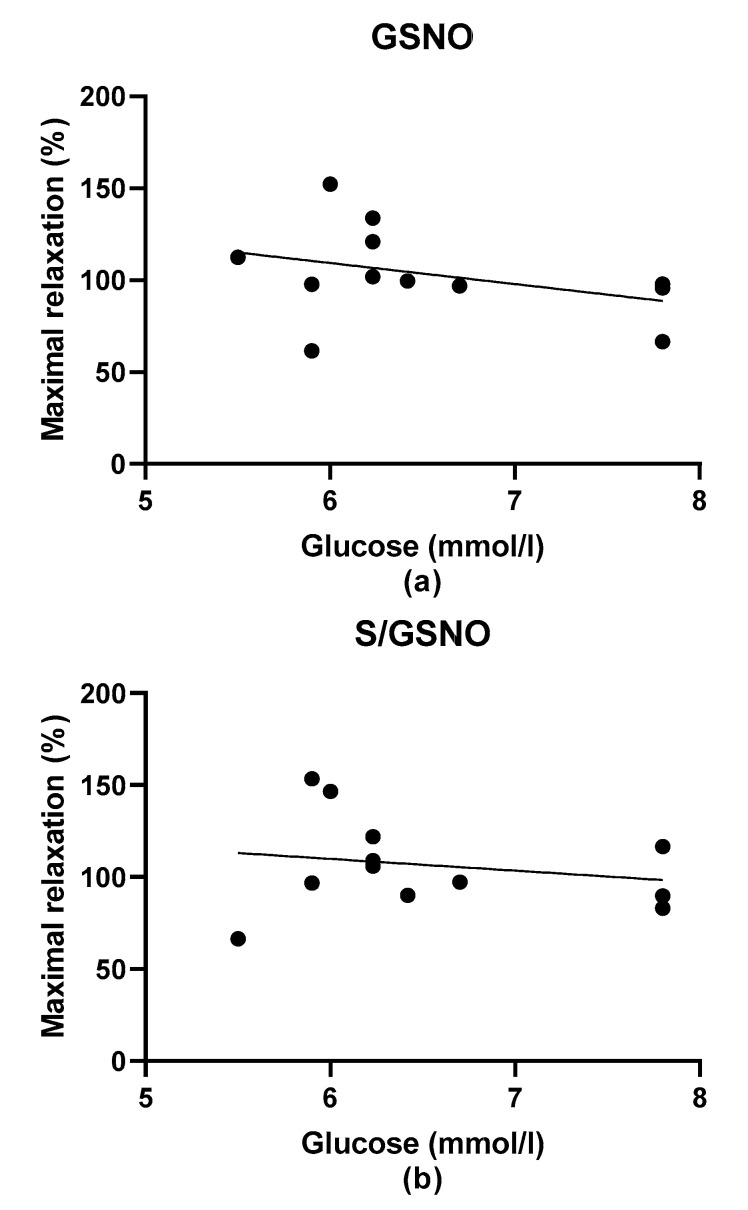
The mutual relationship between plasma glucose levels and the maximal relaxation induced by GSNO (**a**) and S/GSNO products (**b**). There was no dependency in this respect for both GSNO and S/GSNO products.

**Figure 10 molecules-25-02886-f010:**
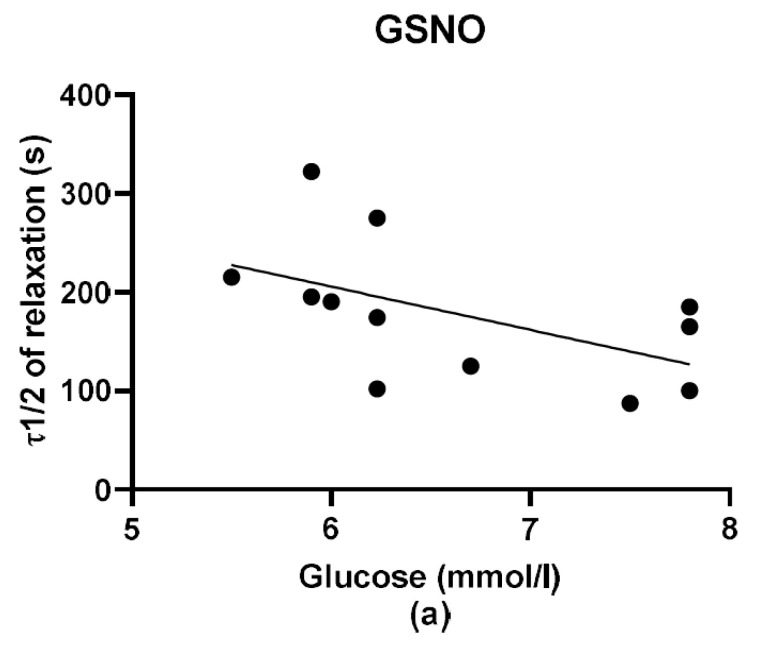
The mutual relationship between plasma glucose levels and the time required to reach 50% of the maximum relaxation (τ1/2) induced by GSNO (**a**) and S/GSNO products (**b**). The speed of relaxation induced by S/GSNO products positively correlated with the glucose level of patients (Y = −36.03*X + 331.2).

**Figure 11 molecules-25-02886-f011:**
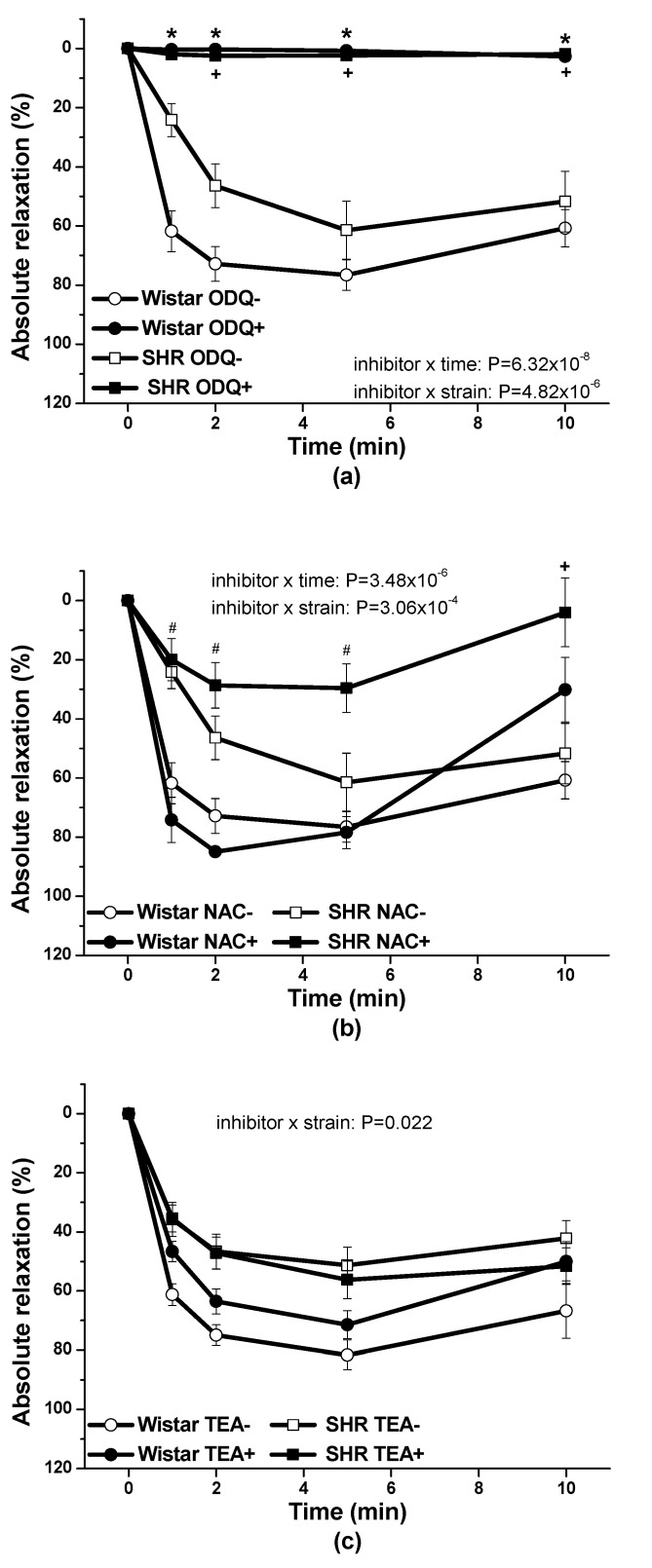
The possible mechanism of the vasorelaxation induced by S/GSNO products in rat thoracic aorta. The comparison of the absolute relaxation induced by S/GSNO products (0.5 µmol/L) before and after the treatment: (**a**) with inhibitor of soluble guanylate cyclase (ODQ; 10 µmol/L); (**b**) HNO scavenger (NAC; 1 mmol/L); and (**c**) non-specific inhibitor of K+ channels (TEA, 1 mmol/L) in normotensive Wistar and spontaneously hypertensive rat (SHR). Results are expressed as the mean ± S.E.M. Significant differences were evaluated by tree-way ANOVA for main factors: used time, inhibitor and strain. Bonferroni post hoc test was used to describe the differences in mean values of the experimental groups. * *p* < 0.05 with respect to the value of the S/GSNO response without inhibitor in Wistar rats within the appropriate graph; + *p* < 0.05 with respect to the value of the S/GSNO response without inhibitor in SHR within the appropriate graph; # *p* < 0.05 with respect to the value of the S/GSNO response with inhibitor in Wistar rats within the appropriate graph.

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
