# Peer review of "Arterial Hypertension and Plasma Glucose Modulate the Vasoactive Effects of Nitroso-Sulfide Coupled Signaling in Human Intrarenal Arteries"

_molecules, 2020, doi:10.3390/molecules25122886_

Round 1
Reviewer 1 Report
In this manuscript Sona Cacanyiova et al. continue to investigate the nitro-sulfide signaling pathway. They evaluated vasoactive effect of the NO donor (GSNO) and the mixture of H2S donor with GSNO (S/GSNO) in intrarenal arteries obtained from hypertensive and normotensive patients. Authors have showed different vasoactive response depending on occurrence of hypertension and adding H2S donor to GSNO. Namely, S/GSNO produced faster (and more pronounced in normotensive patients) vasorelaxation than GSNO alone. In arteries from hypertensive patients the response to GSNO was faster and more pronounced compared to normotensives. Also, response to S/GSNO was faster in hypertensives. Additionally, authors provided valuable mechanistic insight into possible endogenous H2S action on intrarenal arteries as they showed the expression of genes coding H2S-producing enzymes in perivascular adipose tissue and the localization of these enzymes in intrarenal arteries. Currently, the proper management of arterial hypertension remains challenging, mostly due to gaps in understanding the pathophysiology of the disease. Therefore I believe that providing a knowledge about the role of vasoactive signaling of gaseous transmitters in hypertension is of utmost importance.
The manuscript could be improved if the authors answer a few minor concerns:
- Showing the correlation between glucose concentration and the speed of relaxation is very interesting, however, the relationship could be a result of differences in speed of relaxation between hypertensives and normotensives, especially as authors have shown that hypertension was associated with increased plasma glucose. Have authors observed similar correlations within hypertensive and normotensive group as well? If the plasma glucose is a confounder related to hypertension it is rather speculative to suggest its causal role in the acceleration of vasoactive response. Also, it would be interesting to know if authors observed any relationship between glucose level and other parameters regarding vasoactive response (e.g. maximum relaxation, duration of the effect etc.).
- There is a mistake in reporting results of relationship between glucose level and vasoactive response. According to the data shown in Figure 5, the time to 50% relaxation not the speed of relaxation negatively correlated with glucose level. The speed of relaxation positively correlated with glucose level. It should be corrected throughout the manuscript.
- In Discussion lines 362-363 authors stated that the difference in maximum relaxation of S/GSNO products was dependent on the incidence of hypertension. However, their results do not support this conclusion (Fig 6a).
- Using abbreviation for cystathionine γ-lyase is inconsistent, CTH or CSE? It should be corrected.
- Figure legends should be improved as in some cases the abbreviations and double asterisks are not defined. Figure 2 also lacks the description what specifically is showed in panel a) and panel b).
Author Response
Thank you for revising of MS and for your valuable help. The comments were accepted and the missing explanations and additions were incorporated into the MS (see MS Cacanyiova et al. with described changes).
In this manuscript Sona Cacanyiova et al. continue to investigate the nitro-sulfide signaling pathway. They evaluated vasoactive effect of the NO donor (GSNO) and the mixture of H2S donor with GSNO (S/GSNO) in intrarenal arteries obtained from hypertensive and normotensive patients. Authors have showed different vasoactive response depending on occurrence of hypertension and adding H2S donor to GSNO. Namely, S/GSNO produced faster (and more pronounced in normotensive patients) vasorelaxation than GSNO alone. In arteries from hypertensive patients the response to GSNO was faster and more pronounced compared to normotensives. Also, response to S/GSNO was faster in hypertensives. Additionally, authors provided valuable mechanistic insight into possible endogenous H2S action on intrarenal arteries as they showed the expression of genes coding H2S-producing enzymes in perivascular adipose tissue and the localization of these enzymes in intrarenal arteries. Currently, the proper management of arterial hypertension remains challenging, mostly due to gaps in understanding the pathophysiology of the disease. Therefore I believe that providing a knowledge about the role of vasoactive signaling of gaseous transmitters in hypertension is of utmost importance.
The manuscript could be improved if the authors answer a few minor concerns:
- Showing the correlation between glucose concentration and the speed of relaxation is very interesting, however, the relationship could be a result of differences in speed of relaxation between hypertensives and normotensives, especially as authors have shown that hypertension was associated with increased plasma glucose. Have authors observed similar correlations within hypertensive and normotensive group as well?
We evaluated the relationship between the plasma glucose level and the speed of relaxation in all patients together. We agree with the reviewer that to do similar correlation within hypertensive and normotensive group would be interesting, nevertheless, it is needed to elevate the number of patients in particular groups to confirm a significance of any dependency. It could be realized in next experiments.
If the plasma glucose is a confounder related to hypertension it is rather speculative to suggest its causal role in the acceleration of vasoactive response.
We agree with the reviewer, we corrected it:
- 1 L 41, P. 11 L 361, P. 13 L 451, P. 16 L 600
Also, it would be interesting to know if authors observed any relationship between glucose level and other parameters regarding vasoactive response (e.g. maximum relaxation, duration of the effect etc.).
We also evaluted the relationship between the plasma glucose level and the maximal relaxation and we did not find out any dependence. We added it to the MS (see Figure 9). We were not able to evaluate the dependence between glucose level and duration of the effect because we generally finnished the measurement of the response after 10 minutes to avoid fatigue of the tissue.
- There is a mistake in reporting results of relationship between glucose level and vasoactive response. According to the data shown in Figure 5, the time to 50% relaxation not the speed of relaxation negatively correlated with glucose level. The speed of relaxation positively correlated with glucose level. It should be corrected throughout the manuscript.
We thank reviewer for notice. We corrected it in MS.
- 10 L 335, P. 11 L 355, P. 13 L 449
- In Discussion lines 362-363 authors stated that the difference in maximum relaxation of S/GSNO products was dependent on the incidence of hypertension. However, their results do not support this conclusion (Fig 6a).
We corrected it.
- 13 L 428
- Using abbreviation for cystathionine γ-lyase is inconsistent, CTH or CSE? It should be corrected.
We corrected it. We replaced CSE with CTH throughout the manuscript.
- Figure legends should be improved as in some cases the abbreviations and double asterisks are not defined. Figure 2 also lacks the description what specifically is showed in panel a) and panel b).
We corrected it.
Reviewer 2 Report
Major points:
1) The tissues were collected from patients with cancer and this condition was not mentioned or discussed at any time. To what extent are the shown results valid for non-cancer patients considering the variety of inflammatory mediators (mainly cytokines and chemokines) produced by tumor and/or immune cells that certainly affect the vascular response?
2) Before studying the kinetic profiles of the vasodilator activities of the compounds of interest, the authors must show the concentration-response curves and with basis on this, justify the chosen concentrations of the test compounds. For example, will the kinetic responses be equivalent if the concentrations of the compounds of interest are 10-times higher or lower? On the other hand, and considering that the authors support the existence of synergistic effects between GSNO and sulfide, it is not accepted the statement "data not shown" for the latter.
3) Fasting plasma glucose is highly variable (either among patients or for a given patient on a day-by-day basis). Instead, glycated hemoglobin (HbA1c) is representative of the mean plasma glucose concentrations to which the vessels are exposed, and the chosen parameter to define effective diabetes control.
4) It seems that the authors statistically compared the kinetic profiles of vascular relaxation at determined time points. This is not correct considering that the time measurements are not independent (e.g., what happens at t=5min depends on what happened at t=2min). In this way, the authors should either analyze their curves after calculating the respective areas under the curves or by applying a two-way ANOVA approach followed by a proper post-hoc test.
5) How did the authors calculate the time to reach 50% relaxation? They should state which mathematical model of relaxation vs. time curve they applied (exponential decay?, second order polynomial?, etc.) in order to extrapolate the individual T1/2 values. This is a central point considering that in all the cases the authors calculated a T1/2 value from a 3 point curve (relaxation values assessed 60, 120 and 300 min after the addition of the compounds of interest) which means that the extrapolated T1/2 values are affected by a great degree of variability considering a 3-point curve.
6) Considering that the authors speculate about the formation of NO-/HNO from H2S + GSNO, they should test the effects of Angeli's salt in their preparation as well as the involved mechanisms (for example, sGC-cGMP, K(Ca) or K(ATP)-channels, etc.). The participation of ROS should also be evaluated in the differential response of arteries from normotensive and hypertensive patients to the mix H2S-GSNO.
7) Which evidence of "nitrate tolerance" do the authors have in their model to propose the use of the mix H2S-GSNO as a "prospective pharmacological tool"? Can GSNO be considered an organic nitrate that can potentially induce tolerance? The authors should revise these concepts.
Author Response
Thank you for revising of MS and for your valuable help. The comments were accepted and the missing explanations and additions were incorporated into the MS (see MS Cacanyiova et al. with described changes).
Major points:
1) The tissues were collected from patients with cancer and this condition was not mentioned or discussed at any time. To what extent are the shown results valid for non-cancer patients considering the variety of inflammatory mediators (mainly cytokines and chemokines) produced by tumor and/or immune cells that certainly affect the vascular response?
While we have recieved limited amount of vessel tissue after the surgery, we were not able to evalute any inflamatory markers in the tissue. However, we checked the values of CRP, a protein whose circulating concentrations rise in plasma in response to inflammation, in patients. We found out that despite of the fact that there was not difference between CRP value in normotensive (74,98 ± 17.07) and hypertensive (patients 38.58 ± 19,87) we demonstrated the difference in vascular responses between normotensive and hypertensive patients.
Nevertheless, we can not exclude the possible effect of inflammatory mediators produced by tumor and/or immune cells on vascular responses in comparison with non-cancer patients. We added the missing information to the MS.
P.14 L 480-490
2) Before studying the kinetic profiles of the vasodilator activities of the compounds of interest, the authors must show the concentration-response curves and with basis on this, justify the chosen concentrations of the test compounds. For example, will the kinetic responses be equivalent if the concentrations of the compounds of interest are 10-times higher or lower? On the other hand, and considering that the authors support the existence of synergistic effects between GSNO and sulfide, it is not accepted the statement "data not shown" for the latter.
In our previous studies we showed the concentration-response curves for Na2S (see Cacanyiova et al J Physiol Pharmacol 67(4): 501-512, 2016 and Berenyiova et al J Physiol Pharmacol 69, 4, 2018). We confirmed that in human lobar arteries, the Na2S bolus application at a 1 μmol/L concentration had a minimal effect on vascular tone, and a 20 μmol/L concentration evoked approximately 14% vasorelaxation (Cacanyiova et al. 68, 4, 527-538, 2017). To avoid fatigue of the tissue we did not make the concentration-response curves in this experiment.
However, based on results of Cortese-Krott et al. (Redox Biol. 2: 234–244, 2014) and also on our previous results (Berényiová et al. Nitric Oxide 46, 123–130, 2015), for preparation of Na2S-GSNO mixture, which produces new effective products (their formation was followed by UV–VIS spectroscopy, absorbance increase at λmax 412 nm) we strictly needed to keep following conditions:
- to use such Na2S concentration that has minimal or no effect on vascular tone (0.5 µmol/L was suitable),
- to use a 10:1 molar excess of Na2S over GSNO for the preparation of the mixture.
We added this information to the MS:
P.15 L548-549
We removed the statement „data not shown“ and we added results:
P.5 L 203-205
3) Fasting plasma glucose is highly variable (either among patients or for a given patient on a day-by-day basis). Instead, glycated hemoglobin (HbA1c) is representative of the mean plasma glucose concentrations to which the vessels are exposed, and the chosen parameter to define effective diabetes control.
In our study venous blood samples were collected periodically (early morning) during hospitalization of the patient for measurement of fasting plasma glucose. This parameter has already been recognized as the diagnostic test of choice. E.g. the relation between fasting glycemia plasma levels and venous endothelium-dependent dilation in normotensive and hypertensive patients has been evaluated (Rubira et al. J Clin Hypertens. 9:859–865, 2007). We agree that HbA1C is the standard for glycemic control that best correlates with mean plasma glucose. However, targeting fasting plasma glucose could be more beneficial when Hb A1c results are very high, whereas targeting postprandial glucose is more effective when A1C results are lower (Schrot, Clinical Diabetes 22: 169-172, 2004). Nevertheless, we could focus our attention on this parameter in the future.
4) It seems that the authors statistically compared the kinetic profiles of vascular relaxation at determined time points. This is not correct considering that the time measurements are not independent (e.g., what happens at t=5min depends on what happened at t=2min). In this way, the authors should either analyze their curves after calculating the respective areas under the curves or by applying a two-way ANOVA approach followed by a proper post-hoc test.
We agree with the reviewer, we re-calculated the results using two-way ANOVA (see Figures 4-6).
5) How did the authors calculate the time to reach 50% relaxation? They should state which mathematical model of relaxation vs. time curve they applied (exponential decay?, second order polynomial?, etc.) in order to extrapolate the individual T1/2 values. This is a central point considering that in all the cases the authors calculated a T1/2 value from a 3 point curve (relaxation values assessed 60, 120 and 300 min after the addition of the compounds of interest) which means that the extrapolated T1/2 values are affected by a great degree of variability considering a 3-point curve.
For evaluation of the time to reach 50% relaxation we used the original curves (we added an example to the MSD, see Figure 9). Time-dependent relaxation fitted the simple exponential decay function, F = y0 +a*exp(-k*t), where k is the rate constant and for τ 1/2 = ln2/k.
We added this information to the MS
P.8 L 292-295
6) Considering that the authors speculate about the formation of NO-/HNO from H2S + GSNO, they should test the effects of Angeli's salt in their preparation as well as the involved mechanisms (for example, sGC-cGMP, K(Ca) or K(ATP)-channels, etc.). The participation of ROS should also be evaluated in the differential response of arteries from normotensive and hypertensive patients to the mix H2S-GSNO.
We thank reviewer for suggestion. We agree that the evaluation above mentioned pathways would be interesting. We were not able to do it in our experiment because of fatigue of the tissue. Because the acquirement of suitable human subject (and sample) is time consuming we plan incorporate the evaluation of the mechanisms in the next study.
7) Which evidence of "nitrate tolerance" do the authors have in their model to propose the use of the mix H2S-GSNO as a "prospective pharmacological tool"? Can GSNO be considered an organic nitrate that can potentially induce tolerance? The authors should revise these concepts.
In our study we confirmed that the modulation of the vasoactive properties of GSNO by reaction with a H2S donor could probably alleviate the negative effects, such as increased sensitivity and reaction to GSNO as NO donor. The question of the „nitrate tolerance“ was only the object of the discussion. Low molecular weight S-nitrosothiols (such as GSNO) do not induce tolerance and their relaxant effect is retained in glyceryltrinitrate-tolerant arteries (Miller et al., Eur. J. Pharmacol. 408, 335–343, 2000). Andrews et al. (Clin Sci (Lond) 129: 179-87, 2015) confirmed, that HNO as one of the potential end-products of the GSNO and H2S interaction, is also resistant to tolerance development in human blood vessels. We hypothesized that using the mix GSNO-H2S could provide both: to decrease the response of GSNO as NO donor and to keep the resistance to tolerance development. Nevertheless, we revised this part and removed the speculative statement.
P. 14 L 472 -473, 476, 479-480
Reviewer 3 Report
The authors studied the vasoactive effects of nitroso-sulfide compound in the human intrarenal arteries of normal and hypertensive patients. They found that in human lobar arteries from normotensive patients, the nitroso-sulfide (S/GSNO) products induced a more pronounced vasorelaxation compared to the effect induced by GSNO. In human lobar arteries from hypertensive patients, the vasorelaxation induced by S/GSNO products was similar to that observed with GSNO alone. Moreover in hypertensive arteries the vasorelaxation induced by S/GSNO products was faster compared to the vasodilatation observed in normotensive arteries in terms of initiation (1-2 minutes after administration) and return (10 minutes after the administration). Moreover the authors demonstrated that in perivascular adipose tissue CBS and CSE gene expression was significantly higher than in arterial wall in both normotensive and hypertensive arteries. The study is well described however some additional experiments need to be performed to further improve the paper.
- Beyond CBS and CSE, 3MST is the third H2S-produce enzyme. The levels of 3MST as mRNA and protein expression should also be evaluated in arterial wall and perivascular adipose tissue.
- The authors should clarify the discrepancy by gene expression and immunoflorescence technique. In RT-PCR, CSE expression in arterial wall is not detectable meanwhile by immunoflorescence CSE protein has been localized.
- Molecular mechanism by which S/GSNO potentiates vasorelaxant effect of GSNO has not been elucitated. One hypothesis could be due to the antioxidant action induced by Hydrogen sulphide. The other possibility could be the inhibition of phosphodiesterases, one main target of hydrogen sulphide (Bucci et al., 2010). In particular by inhibiting fosfodiesterases, the H2S increases the half-life of cGMP, the product of sGC activation induced by NO. The authors should performed some experiments in order to clarify these issues.
Minor Comments
Figure 1 is confunded. The relative gene expression could be divided in normal and hypertensive arteries. To put all the data together is not clear
In the legend of figure 1 the data are expressed as mean ± SE meanwhile in the legend of figure 2 the data are expressed as mean ± SEM, the authors should use the same expression in all the legends.
In figure 4a, the relaxation induced by S/GSNO exceed the 100% of relaxation after 5 minutes, relaxation could not be more than 100%
In figure 4b there is no SEM in S/GSNO curve after 5 minutes
Author Response
Thank you for revising of MS and for your valuable help. The comments were accepted and the missing explanations and additions were incorporated into the MS (see MS Cacanyiova et al. with described changes).
The authors studied the vasoactive effects of nitroso-sulfide compound in the human intrarenal arteries of normal and hypertensive patients. They found that in human lobar arteries from normotensive patients, the nitroso-sulfide (S/GSNO) products induced a more pronounced vasorelaxation compared to the effect induced by GSNO. In human lobar arteries from hypertensive patients, the vasorelaxation induced by S/GSNO products was similar to that observed with GSNO alone. Moreover in hypertensive arteries the vasorelaxation induced by S/GSNO products was faster compared to the vasodilatation observed in normotensive arteries in terms of initiation (1-2 minutes after administration) and return (10 minutes after the administration). Moreover the authors demonstrated that in perivascular adipose tissue CBS and CSE gene expression was significantly higher than in arterial wall in both normotensive and hypertensive arteries. The study is well described however some additional experiments need to be performed to further improve the paper.
Beyond CBS and CSE, 3MST is the third H2S-produce enzyme. The levels of 3MST as mRNA and protein expression should also be evaluated in arterial wall and perivascular adipose tissue.
We agree, the levels of 3MST as mRNA and protein expression should also be evaluated in arterial wall and perivascular adipose tissue. In our preliminary measurements we were not able to detect this enzyme, however, we can add such evaluation to our next series of experiments.
The authors should clarify the discrepancy by gene expression and immunoflorescence technique. In RT-PCR, CSE expression in arterial wall is not detectable meanwhile by immunoflorescence CSE protein has been localized.
The absence of CSE RNA in arterial wall might be explained by extremely high metabolic turnover of the specific mRNA. This hypothesis is supported by the presence of CSE protein only in seleted cells of arterial wall. This fact - hypothesis however needs further investigation.
We added this information to the discussion.
P.11 L 380, P. 12 L 381-383
Molecular mechanism by which S/GSNO potentiates vasorelaxant effect of GSNO has not been elucitated. One hypothesis could be due to the antioxidant action induced by Hydrogen sulphide. The other possibility could be the inhibition of phosphodiesterases, one main target of hydrogen sulphide (Bucci et al., 2010). In particular by inhibiting fosfodiesterases, the H2S increases the half-life of cGMP, the product of sGC activation induced by NO. The authors should performed some experiments in order to clarify these issues.
We thank reviewer for suggestion. We agree that the evaluation above mentioned pathways would be interesting. We were not able to do it in our experiment because of fatigue of the tissue. Because the acquirement of suitable human subject (and sample) is time consuming we plan incorporate the evaluation of the mechanisms in the next study. Nevertheless we added the information about the next possible mechanisms of S/GSNO action in the discussion.
P.13 L 453-459.
Round 2
Reviewer 2 Report
All the points raised on the original version of the manuscript were appropriately answered in the present version.
Author Response
We thank the reviewer for answer.
Reviewer 3 Report
Dear authors,
in my opinion molecular mechanism by which S/GSNO potentiates vasorelaxant effect of GSNO needs to be added to the manuscript.
I understand the difficulty to obtain other human tissues but this experiment improve the manuscript.
Author Response
Dear reviewer.
We added to the text information related to the possible mechanism by which S/GSNO could potentiate vasorelaxant effect of GSNO. Some of them resulted from our previous experiments, which have already been published (see Berenyiova et al. 2015, 2020). The remaining part resulted from our preliminary experiments using rat samples (normotensive Wistar rats and spontaneously hypertensive rats). Originally, we planned it use for next publication. Nevertheless, we agree with the reviewer that such results can improve the manuscript. So we added it to the MS. To evaluate the role of K+ channels in vasorelaxant effect of S/GSNO we used tetraethylammonium chloride (TEA) - non-selective K+ channel blocker and we also used ODQ to evaluate the role of soluble guanylate cyclase and N-acetylcysteine to evaluate the role of HNO as a mediator.
Lines: 37-39, 341-363, 373-383 (Figure 11), 431-433, 494-514, 608-634, 662-663, 675-677.